# Mortality in ICU Patients with COVID-19-Associated Pulmonary Aspergillosis

**DOI:** 10.3390/jof9060689

**Published:** 2023-06-20

**Authors:** Anna Beltrame, David A. Stevens, Donna Haiduven

**Affiliations:** 1College of Public Health, University of South Florida, Tampa, FL 33622, USA; dhaiduve@usf.edu; 2California Institute for Medical Research, San Jose, CA 95128, USA; stevens@stanford.edu; 3Division of Infectious Diseases and Geographic Medicine, Stanford University Medical School, Stanford, CA 94305, USA

**Keywords:** invasive aspergillosis, CAPA, COVID-19, SARS-CoV-2, outcome, ICU, mortality

## Abstract

A review of 38 studies involving 1437 COVID-19 patients admitted to intensive care units (ICUs) with pulmonary aspergillosis (CAPA) was conducted to investigate whether mortality has improved since the pandemic’s onset. The study found that the median ICU mortality was 56.8%, ranging from 30% to 91.8%. These rates were higher for patients admitted during 2020–2021 (61.4%) compared to 2020 (52.3%), and prospective studies found higher ICU mortality (64.7%) than retrospective ones (56.4%). The studies were conducted in various countries and used different criteria to define CAPA. The percentage of patients who received antifungal therapy varied across studies. These results indicate that the mortality rate among CAPA patients is a growing concern, mainly since there has been an overall reduction in mortality among COVID-19 patients. Urgent action is needed to improve prevention and management strategies for CAPA, and additional research is needed to identify optimal treatment strategies to reduce mortality rates among these patients. This study serves as a call to action for healthcare professionals and policymakers to prioritize CAPA, a serious and potentially life-threatening complication of COVID-19.

## 1. Introduction

The epidemiology of invasive pulmonary aspergillosis (IPA), mainly caused by members of the ubiquitous *Aspergillus fumigatus complex*, has changed [1]. Until recently, IPA was considered an opportunistic disease exclusive to immunosuppressed patients receiving chemotherapy, transplantation, and small molecule kinase inhibitors [2]. Standardized ventilation protocols have been implemented for these patients to reduce exposure to Aspergillus spores and IPA risk [2]. However, IPA has been frequently linked to patients requiring ventilatory support in intensive care units (ICUs) [3]. The risk factors for developing IPA in ICU patients include chronic obstructive pulmonary disease (COPD), kidney disease, cirrhosis of the liver, HIV infection, and diabetes [4]. New studies have shown that IPA also occurs in patients with viral respiratory diseases requiring intensive care and often leading to death, such as influenza and coronavirus disease (COVID-19) [5,6]. There are estimates that up to 20% of influenza patients admitted to ICUs may develop influenza-associated pulmonary aspergillosis [6]. This is a significant public health concern given the high number of influenza and COVID-19 cases currently occurring worldwide [7,8].

IPA has been described as a complication in critically ill patients hospitalized for COVID-19 and is defined as COVID-19-associated pulmonary aspergillosis (CAPA) [5]. A qualitative review of 41 studies conducted worldwide included 6193 hospitalized COVID-19 patients and showed a CAPA incidence rate of 10.9%, ranging from 3.3% in Spain to 30.8% in China [5]. The rate was 11.1% when only considering patients admitted to intensive care units (ICUs) (95.3%), rising to 15.1% once more precise prospective observational studies (*n* = 18) were considered [5]. However, as physician awareness of IPA development risk in ICU patients influences CAPA incidence, the actual rate may be higher [9]. In a prospective multicenter study conducted in Italy, the CAPA incidence reached 27.7% in intubated patients (*n* = 108) who first underwent a fungal screening protocol upon ICU admission, and again seven days later [9]. The same authors showed that the odds of death in intubated patients with CAPA were more than three times the odds in COVID-19 patients without IPA (OR: 3.53; 95% CI: 1.29–9.67), even after adjustment for confounders (age, need for renal replacement therapy, and SOFA score at ICU admission) [9].

To date, the literature on CAPA outcomes occurring in COVID-19 patients during the first period of the COVID-19 pandemic includes one systematic review and meta-analysis focusing on the definition of CAPA [10], one systematic review of the incidence, diagnosis, and outcomes of CAPA conducted analyzing 19 cohort studies [11], and a meta-analysis on mortality that included 20 studies [12]. In studies conducted during 2020, the pooled mortality in CAPA patients was 51.2% (95% CI: 43.1–61.1, I^2^ = 38%), with the risk of death in CAPA patients being 1.84 times (RR 1.84, 95% CI: 1.45–2.33) the risk in COVID-19 patients without IPA [12].

However, since this new viral disease emerged, the clinical outcomes of hospitalized patients who develop CAPA may have changed. The chaotic situation in hospitals at the beginning of the pandemic may have impacted the mortality of CAPA patients, alongside the absence of a SARS-CoV-2 vaccine, an antiviral treatment, and COVID-19 patient management guidelines [13]. The purpose of the present review is to describe all the existing evidence on the mortality of patients admitted to ICUs with CAPA from the beginning of the COVID-19 pandemic to the end of 2022. This is the first step toward understanding whether the fungus plays a key role in CAPA patient death, which would require new prospective research on hospital fungal prevention, as well as antifungal prophylaxis and treatment of COVID-19 patients.

## 2. Materials and Methods

### 2.1. Search Strategy and Study Selection

Peer-reviewed journal papers were included if they:Were observational studies (retrospective or prospective);Assessed 10 or more patients hospitalized in an ICU with CAPA;Obtained the COVID-19 diagnosis through a positive SARS-CoV-2 reverse transcriptase-polymerase chain reaction (PCR) from nasal and/or pharyngeal swabs;Described well-established diagnostic criteria for the CAPA diagnosis as reported in the article by Koehler et al. [13];Described the mortality of CAPA patients (quantitative method studies).

Articles were included regardless of the original language used. The preferred reporting items for systematic reviews and meta-analysis extension for scoping reviews (PRISMA-ScR) checklist was used to conduct a literature search on 12 December 2022 [14]. MEDLINE and EMBASE databases were used, and the results were exported to EndNote. The search keywords “invasive pulmonary aspergillosis” or “COVID-19-associated pulmonary aspergillosis” and “COVID-19” or “severe acute respiratory syndrome coronavirus 2 (SARS-CoV-2)” were combined with Boolean operators. 

### 2.2. Data Extraction and Analysis

Data extraction included the following items: first author and publication year, country of study, study period (enrollment), study design (retrospective or prospective), sample size (number of patients), patient age (mean or median), fungal diagnostic method for CAPA definition, CAPA definition (proven, probable, putative, and possible), ICU admission, invasive mechanical ventilation, treatment type (remdesivir, corticosteroids, antifungals, or tocilizumab), hospital and ICU length of stay (LOS), mortality (ICU mortality, 30-day mortality, 42-day mortality, or 90-day mortality).

The characteristics of studies meeting the inclusion criteria and the quantitative and qualitative descriptive statistics of patients were collected in Microsoft Excel, summarized in tables, and described narratively.

The primary outcomes were ICU mortality among COVID-19 patients with CAPA. We calculated the median and interquartile range (IQR) of mortality obtained from 28 studies. Studies reporting 30-day mortality, 42-day mortality, and 90-day mortality were excluded.

## 3. Results

We found 617 records in databases and two additional records were identified through an examination of the reference lists of relevant articles. After duplicate removal, we screened 402 records, from which we retrieved and reviewed 113 full-text documents. Of these, 75 were excluded for the following reasons: 36 were case reports and series; 16 were reviews, editorials, or letters; 5 were conference abstracts; 5 were not found; 3 did not separate CAPA from other fungal infections; 3 included non-ICU patients; 3 did not quantify the mortality; 2 had unclear patient information; and 2 did not report the diagnostic criteria. The remaining 38 records were considered, for a total of 1437 patients (Figure 1). Next, we searched for documents that cited any of the initially included studies. However, no extra articles that fulfilled the inclusion criteria were found in these searches.

The characteristics of the 38 studies (1437 patients) included in the review are reported in Table 1. These studies were conducted in various countries and were published between 2020 and 2023. Most of the studies were conducted in Europe (*n* = 22). A total of 18 studies were conducted during 2020, 2 during 2021, and 19 studies included patients admitted to ICUs from 2020 to 2021. A total of 22 studies were retrospective cohorts, 13 were prospective, and 3 were both. The sample size (number of patients in each study) varied from 10 to 109. The studies included a range of patients with different ages (Table 1) and comorbidities (Table 2). The definitions of CAPA were based on different criteria, including ECMM/ISHAM (*n* = 30), EORTC/MSGERC (*n* = 3), AspICU (*n* = 2), Modified AspICU (*n* = 2), and Verweij criteria (*n* = 1) (Appendix A).

Table 2 presents information on the percentage of patients who received invasive mechanical ventilation (IMV) and other treatments (antiviral, antifungal, corticosteroids, and tocilizumab), the mortality (ICU mortality, 30-day, 42-day, and 90-day mortality), and the hospital and ICU LOS (reported in days). The percentage of patients who received IMV ranged from 35.6 to 100%. Those who received antifungal therapy ranged from 23 to 100%.

The median ICU mortality in 1141 CAPA patients from 28 studies was 56.8% (IQR 23.1), ranging from 30 to 91.8% (Figure 2). The median mortality was 52.3% (IQR 23.75) if patients were admitted in 2020 (*n* = 12) and 61.4% (IQR 22.6) during the period 2020–2021 (*n* = 16) (Figure 2). In 6 prospective studies, the median mortality rate was 64.7% (IQR 28.9), ranging from 36% [15] to 91.8% [16], whereas in 19 retrospective studies, it was 56.4% (IQR 26.3), ranging from 30% [17] to 76.5% [18]. In three retrospective and prospective studies, the median mortality was 52% (IQR 14.5) (Figure 2).

**Table 1 jof-09-00689-t001:** Characteristics of the 38 studies included in the review.

Author [Reference Number], Publication Year	Study Location	Enrolment	Study Design	Patients *n*	Age, Years Mean ± SD or Median (IQR)	CAPA Definition	Proven/Probable/Putative/Possible %
Araya-Rojas [19], 2021	Chile	5/2020 4/2021	R	11	56 ± 10	ECMM/ISHAM	0/100/0/0
Bartoletti [9], 2021	Italy	2/2020 4/2020	P	30	63 (57–70)	Verweij criteria	0/100/0/0
Bentvelsen [20], 2022	The Netherlands	3/2020 4/2020	R	58	69 (60–74)	ECMM/ISHAM	0/50/0/50
Bretagne [21], 2021	France	2/2020 5/2020	R	154	66 ± 9.7	EORTC/MSGERC	NA
Calderon-Parra [22], 2022	Spain	3/2020 8/2021	R	28	68 (65–72)	ECMM/ISHAM	0/57.1/0/42.9
Casalini [23], 2022	Italy	8/2020 5/2021	R	20	66 (60–72)	ECMM/ISHAM	0/100/0/0
De Almeida [24], 2022	Brazil	4/2020 7/2021	R	14	70.4 ± 8.5	ECMM/ISHAM	0/100/0/0
Delliere [25], 2020	France	3/2020 5/2020	R	21	63 (56.8–68.3)	EORTC/MSGERC	0/100/0/0
Dupont [26], 2021	France	3/2020 4/2020	P	19	70 ± 10.5	AspICU	0/0/100/0
Er [27], 2022	Turkey	11/2020 4/2021	P	43	68.5 ± 12.5	ECMM/ISHAM	0/60.5/0/39.5
Erami [28], 2022	Iran	8/2020 6/2021	R	17	77 ± 18, 73.8 (45–88)	ECMM/ISHAM	0/100/0/0
Ergun [29], 2021	The Netherlands, Belgium, France, and UK	2/2020 5/2020	P	39	65 (58–75)	ECMM/ISHAM	2.6/97.4/0/0
Fischer [30], 2022	Switzerland	3/2020 3/2021	P	13	70.3 ± 7.8	ECMM/ISHAM	7.7/76.9/0/15.4
Fortun [31], 2023	Spain	3/2020 6/2021	R	108	65.5 ± 12.1	ECMM/ISHAM	100 pr or pb
Gangneux [32], 2022	France	2/2020 7/2020	R&P	76	63.3 ± 12.5	ECMM/ISHAM	100 pr or pb
Giacobbe [33], 2022	Austria, Italy, Germany, UK, and Belgium	3/2020 4/2021	P	56	NA	ECMM/ISHAM	9/91/0/0
Giusiano [34], 2022	Argentina	3/2020 10/2020	P	19	65 ± 8.6	ECMM/ISHAM	0/100/0/0
Hashim [35], 2022	India	3/2020 8/2021	R&P	74	55 (44.8–64.3)	ECMM/ISHAM	2.7/68.9/0/28.4
Hatzl [36], 2021	Austria	9/2020 5/2021	P	10	NA	ECMM/ISHAM	0/90/0/10
Huang [37], 2022	Taiwan	5/2021 8/2021	R	11	71 (62–77)	ECMM/ISHAM	0/90.9/0/9.1
Iqbal [16], 2021	Pakistan	6/2020 5/2021	P	61	60.7 ± 8.7	ECMM/ISHAM	0/100/0/0
Janssen [38], 2021	The Netherlands, Belgium, and France	2/2020 5/2020	R&P	42	68 (61–73)	ECMM/ISHAM	14.3/76.2/0/9.5
Kim [39], 2022	The Republic of Korea	7/2020 3/2021	R	17	73 (70–77)	ECMM/ISHAM	0/88.2/0/11.8
Koukaki [40], 2022	Greece	8/2020 11/2021	R	14	48 (43–70)	ECMM/ISHAM	7.1/57.1/0/35.8
Lahmer [15], 2021	Germany	3/2020 4/2020	P	11	72 (58–84)	Modified AspICU	0/0/100/0
Lee [41], 2022	The Republic of Korea	1/2020 5/2021	R	10	71.5 (64–77)	ECMM/ISHAM	NA
Leistner [42], 2022	Germany	1/2020 12/2020	R	47	67.4 (62.4–75.9)	ECMM/ISHAM	4.3/61.7/0/34
Marta [43], 2022	Spain	3/2020 12/2020	P	35	68.8 ± 8.1	ECMM/ISHAM	0/20/0/80
Melchers [44], 2022	The Netherlands	1/2021 7/2021	R	13	68 ± 7	ECMM/ISHAM	23/77/0/0
Permpalung [45], 2022	The USA	3/2020 8/2020	R	39	66 (55–70)	ECMM/ISHAM	0/51.3/0/48.7
Prattes [46], 2022	Austria, Belgium, France, Germany, Italy, Pakistan, Spain, the UK, and the USA	3/2020 5/2021	P	109	68 (60–75)	ECMM/ISHAM	10.1/73.4/0/16.5
Ranhel [17], 2021	Portugal	11/2020 2/2021	R	10	65.8 ± 8.6	ECMM/ISHAM	0/60/0/40
Rouze [47], 2022	France, Spain, Greece, Portugal, and Ireland	2/2020 5/2020	R	14	67 (52–75)	AspICU	0/0/100/0
Sivasubramanian [18], 2021	The USA	1/2020 3/2021	R	48	67 (49–86)	ECMM/ISHAM	4.2/18.8/0/77
Velez Pintado [48], 2021	Mexico	3/2020 7/2020	R	16	64 ± 10	ECMM/ISHAM	12.5/87.5/0/0
White [49], 2020	The UK	2020	P	19	NA	Modified AspICU, own CAPA definition	NA
Xu [50], 2021	China	12/2019 4/2020	R	78	64.3 ± 13.6	EORTC/MSGERC	NA
Zhang [51], 2021	The USA	3/2020 8/2020	R	33	63.2 (38–85)	ECMM/ISHAM	0/48.5/0/51.5

R = retrospective; P = prospective; R&P = retrospective and prospective; NA = not available.

**Table 2 jof-09-00689-t002:** Characteristics of 38 studies included in the review and mortality reported in CAPA patients.

Author = [Reference Number], Publication Year	IMV %	R %	Antifungal Therapy %	C %	T %	H and ICU LOS, Days, Mean ± SD or Median (IQR)	Patients Who Died *n* (%) #
Araya-Rojas [19], 2021	100	0	100	100	0	NA	4 (36.4)
Bartoletti [9], 2021	100	10	53	60	73	16 ± 13.3 ICU	13 (44 *)
Bentvelsen [20], 2022	100	0	72.4	29	0	NA	23 (39.7 *)
Bretagne [21], 2021	100	0	77.3	31.8	0	26 (16–36)	71 (46.1)
Calderon-Parra [22], 2022	100	14.3	96.4	100	92.9	66 (43–88) H 57 (28–85) ICU	17 (60.7)
Casalini [23], 2022	100	0	70	90	0	NA	13 (65)
De Almeida [24], 2022	100	0	84.7	92.9	0	NA	10 (71.4)
Delliere [25], 2020	95.2	0	NA	28.6	9.5	21.05 ± 17.6	15 (71.4)
Dupon [26], 2021	100	0	47.4	5.3	0	NA	7 (36.8 **)
Er [27], 2022	88.4	2.3	39.5	92.9	0	29 (19–41) H 23 (13–40) ICU	29 (67.4)
Erami [28], 2022	100	NA	100	35.3	NA	NA	13 (76.5)
Ergun [29], 2021	NA	2.6	71.8	25.6	0	18 (13–30) ICU	21 (53.8 *)
Fischer [30], 2022	NA	7.7	NA	100	NA	NA	8 (62)
Fortun [31], 2023	73.1	23.4	100	7.4	29	35 ± 25 H 20 ± 20 ICU	44 (40.7)
Gangneux [32], 2022	100	5	76	46	0	27 ± 11.9 ICU	47 (61.8)
Giacobbe [33], 2022	NA	NA	NA	NA	NA	NA	30 (54 ***)
Giusiano [34], 2022	NA	NA	73.7	NA	NA	29 ± 20 ICU	8 (42.1)
Hashim [35], 2022	35.6	74.3	70.3	56.8	8.1	18 (12.8–29) H	35 (47.3)
Hatzl [36], 2021	80	NA	100	NA	NA	NA	8 (80 *)
Huang [37], 2022	100	54.5	72.7	100	63.6	NA	6 (55)
Iqbal [16], 2021	100	57.4	100	100	54.1	11 (4–14) ICU	56 (91.8)
Janssen [38], 2021	98	NA	NA	NA	NA	18 (12–27) ICU	22 (52)
Kim [39], 2022	76.5	82.4	94.1	94.1	0	NA	6 (36.3 *) 9 (54.3 ***)
Koukaki [40], 2022	NA	NA	100	71.4	42.9	NA	8 (57.1)
Lahmer [15], 2021	100	NA	100	NA	NA	21 ± 14 ICU	4 (36)
Lee [41], 2022	60	NA	100	100	NA	23 (16–37) H	5 (50)
Leistner [42], 2022	100	NA	23	87.2	data	33 (19–53) H 24 (17–43) ICU	30 (63.8)
Marta [43], 2022	94.3	NA	NA	85.7	57.1	38.8 ± 17.1 H 26.4 ± 15.9 ICU	11 (40 ***)
Melchers [44], 2022	100	NA	NA#	100	100	40 (23–58) H 29 (17–41) ICU	5 (38 ***)
Permpalung [45], 2022	100	23.1	48.7	66.7	23.1	41.1 (20.5–72.4)	22 (56.4)
Prattes [46], 2022	88.1	NA	90.7	62.4	14.4	27 (17–42) ICU	77 (71)
Ranhel [17], 2021	100	90	80	50	0	NA	3 (30)
Rouze [47], 2022	100	NA	78.6	71.4	NA	25 (19–28) ICU	5 (35.7 *)
Sivasubramanian [18], 2021	100	60	44	93	0	30 H 23 ICU	40 (83)
Velez Pintado [48], 2021	100	NA	NA	13	75	NA	5 (31)
White [49], 2020	73.7	NA	79	73.7	NA	NA	11 (57.9 *)
Xu [50], 2021	57.7	NA	NA	NA	3.9	21 (15–33) H 17 (10–29) ICU	41 (52.6)
Zhang [51], 2021	NA	NA	61	NA	NA	NA	22 (67)

R = remdesivir; C = corticosteroids; T = tocilizumab; NA = not available; ICU = intensive care unit; IMV = invasive mechanical ventilation; H = hospital, LOS = length of stay; # number of patients who died during the ICU stay or ICU mortality, * 30 day-mortality, ** 42 day-mortality, and *** 90 day-mortality, #23% of patients received antifungal prophylaxis.

## 4. Discussion

During the first wave of COVID-19, the mortality for COVID-19 in ICU patients declined significantly from 50% to 40% from March to May 2020. This was due to a better understanding of the disease and greater experience in ICU management [52]. On 22 June 2020, the University of Oxford released its preliminary results of the Randomized Evaluation of COVID-19 Therapy (RECOVERY) trial in a preprint paper, showing that dexamethasone 6 mg once daily for up to 10 days was able to reduce the 28-day mortality rate in either ventilated COVID-19 patients or those requiring oxygen therapy [53]. The study demonstrated that patients receiving routine care had a 28-day mortality rate of 41% if they received mechanical ventilation, 25% if treated with oxygen only, and 13% if they received no respiratory treatment [53]. The use of dexamethasone reduced deaths by a third in COVID-19 patients requiring mechanical ventilation (RR 0.65 (95% CI 0.48–0.88); *p* = 0.0003) and by a fifth in those receiving oxygen (0.80 (95% CI 0.67–0.96; *p* = 0.0021), thus becoming a standard of care in the management of patients with severe COVID-19 [54]. This improvement in care was then reinforced with remdesivir in severe COVID-19 patients thanks to the results of the Adaptive COVID-19 Treatment Trial-1 (ACTT-1), which demonstrated that the antiviral drug was superior to placebo in improving time to recovery [55,56,57]. In particular, clinical improvement rose faster with remdesivir, although it did not reduce time to death [58]. In 2021, a study demonstrated that if using tocilizumab, the anti-interleukin-6 receptor monoclonal antibody, the outcomes of patients with severe COVID-19 improved [59]. Baricitinib, when combined with remdesivir therapy, has consistently shown a reduction in mortality among hospitalized COVID-19 patients, supported by studies such as ACTT-2, COV-BARRIER, and RECOVERY [60]. A recent meta-analysis of four studies also demonstrated a significant decrease in 28-day mortality when baricitinib was administered with dexamethasone and/or anti-IL6 inhibitors (OR 0.69, 95% CI 0.50–0.94) [60]. Baricitinib and tocilizumab are now FDA-approved for treating severe hospitalized COVID-19 patients. Finally, the PANAMO trial suggests that vilobelimab shows promise as a therapeutic drug, with an absolute risk reduction of 11.2% (95% CI 1.4–21.0%) observed in the analysis for all-cause mortality at day 28 [61]. Additionally, widespread vaccination campaigns have helped protect vulnerable populations and reduce the overall number of severe cases requiring ICU admission. This global improvement in managing ICU-admitted COVID-19 patients permitted a reduction in deaths [62]. In the last meta-analysis, including about 1 million patients with COVID-19 admitted to ICUs, the case fatality rate (CFR) due to COVID-19 was 37.3% (95% CI: 34.6–40.1) [62].

Patients hospitalized in an ICU with severe COVID-19 are at increased risk of developing secondary infections, such as invasive fungal infections [63,64]. This is because they have underlying medical conditions, such as chronic respiratory diseases, and are often treated with corticosteroid therapy and intubation/mechanical ventilation [63,64]. Although there are many reasons for death in critically ill patients with COVID-19, the role of co-infections and their significance are not well understood, with limited research done in this area. A systematic review and meta-analysis showed that 19% of patients hospitalized for COVID-19 had co-infections, and 24% had superinfections [65]. The odds of death were higher among co-infected patients (OR = 2.84; 95% CI: 1.42–5.66) and superinfected patients (OR = 3.54; 95% CI: 1.46–8.58) [65]. *Aspergillus* spp. were the most frequently reported fungi among co-infections (6.7%) and superinfections (13.5%), followed by *Candida* (1% and 18.8%), and *Mucorales* (0.3 and 0.2%) [65]. SARS-CoV-2 may cause damage to the bronchial mucosa and result in injury to the alveoli, which can increase pulmonary epithelial and vascular permeability [66]. This environment may create conditions that are favorable for the invasion of *Aspergillus* spp. (which causes IPA), such as blunting the immunological response, and the use of broad-spectrum antibacterials [64]. The co-occurrence of viral infection and fungal invasion can then lead to more severe illness and complications in COVID-19 patients [66].

Koehler and other international experts have proposed a set of criteria to define CAPA and issued guidelines for its diagnosis and management [13]. Despite increasing reports of CAPA over time, the exact incidence of this complication in ICU patients with COVID-19 remains uncertain. Studies have reported an overall incidence of CAPA in ICU patients of 11.1%, with variation depending on study design, geographical location, the diagnostic tests used, and the CAPA definition [5]. Retrospective and partially prospective studies (*n* = 21) reported a lower incidence rate of 7.1%, whereas solely prospective studies (*n* = 18) reported a higher incidence rate of 15.1% [5]. A geographic variation in incidence rates was also observed, ranging from 3.3% in Greece to 38.7% in Germany [5]. Incidence rates also depended on diagnostic tests and the CAPA definition used [5]. During the first wave of COVID-19, the restricted use of bronchoscopies in critically ill patients posed challenges to the diagnosis of potential CAPA patients, due to the risk to healthcare personnel and the overwhelming number of patients [67]. Additionally, many laboratories stopped manipulating at-risk respiratory samples or determining galactomannan in respiratory samples, likely due to concerns about the transmission of COVID-19 [68]. Moreover, many published cases of CAPA have been diagnosed without patients meeting the correct diagnostic criteria [69]. Fekkar et al. applied the criteria proposed by Koehler et al. to the studies published, and the incidence of proven/probable CAPA fell to 6.1% [70]. Finally, environmental factors, such as hospital construction, maintenance, demolition, and renovation projects that do not adhere to rigorous ventilation requirements may release fungal spores into the air, leading to increased concentrations of these organisms in the surrounding environment [71].

The incidence of mortality among critically ill patients with CAPA also depends on the factors that influence its incidence rate. Studies conducted in 2020 and included in a systematic review show that the pooled mortality for CAPA patients was 51.2% (95% CI: 43.1–61.1, I^2^ = 38%), ranging from 49.4% in prospective studies to 57.2% in retrospective studies [12]. Having defined risk factors or being treated with antivirals and antifungals did not change the proportion significantly [12]. The odds of death in CAPA patients were 2.83 (95% CI: 1.8–4.5) times the odds for COVID-19 patients without IPA. As the analysis included only 20 studies (215 patients), each describing between 4 and 21 patients, the small sample size represents the most important limitation of this study [12]. By comparison, our review, which included 38 studies and showed the characteristics of 1437 CAPA patients admitted to ICUs from 2020 to 2021, found a median ICU mortality of 56.8%, ranging from 30% [17] to 91.8% [16]. The wide discrepancy in mortality between the studies analyzed may be associated with management differences in severe COVID-19 patients among countries and centers. The highest mortality was reported in Pakistan by Iqbal et al. [16] with the results of a prospective study conducted from June 2020 to May 2021. The reported mortality in CAPA patients was 91.8%, although all 61 patients were treated with voriconazole and corticosteroids, and 57.4 and 54.1% of them received remdesivir and tocilizumab, respectively. No association between mortality and various factors was explored, preventing the formation of hypotheses. However, the authors reported that 70.5% of CAPA patients developed septic shock and required inotropic support [16]. The lowest mortality was reported in Portugal by Ranhel et al. [17], who conducted a retrospective study that collected data from 10 patients admitted to ICUs from November to February 2021. A high rate of remdesivir use (90%) and voriconazole treatment (80%) were reported, whereas corticosteroids were used in 50% of the patients. A probable and possible CAPA diagnosis following the ECMM/ISHAM definition was obtained in 60% and 40% of the cases, respectively.

Although there has been an improvement in the mortality of ICU patients with COVID-19 over time, our analysis suggests that this trend is not as apparent in those who develop CAPA. Our review identified a difference in the median of ICU mortality between studies reporting data from patients admitted in 2020 (52.3%) and during the 2020–2021 period (61.4%). Specifically, prospective studies showed higher median mortality (64.7%) compared to retrospective studies (56.4%). These differences could be due to other factors. As awareness of CAPA has increased, more cases may be diagnosed, including those that may have been missed earlier in the pandemic [72]. This could lead to a change in reported mortality. Despite increased awareness, a CAPA diagnosis can still be delayed due to a lack of specific symptoms or radiological patterns, or difficulty in obtaining appropriate samples for testing [68]. Delayed diagnosis could result in more severe cases and higher mortality [12].

The management of severe COVID-19 involves the use of glucocorticoids and other immunosuppressive therapies to improve outcomes [62]. However, while these treatments can be effective, they also come with an increased risk of developing CAPA in patients with severe COVID-19 [73]. This is because these therapies can weaken the immune system, making individuals more susceptible to invasive fungal infections. Although dexamethasone for 10 days reduces mortality in severe COVID-19, its prolonged use is considered a risk factor for developing CAPA [54,73]. Shah et al. [73] showed that COVID-19 patients treated for more than 10 days with corticosteroid therapy had a significantly higher incidence of CAPA than those treated for fewer days (11.9% vs. 4.1%, *p* = 0.0156). In this study’s multivariable analysis, steroid use for more than 10 days was independently associated with CAPA [OR 3.17 (95% CI, 1.02–9.83)] [73].

Some studies show that early antifungal treatment for CAPA patients significantly reduces the risk of mortality, whereas others do not support the same results. After adjusting for confounding factors, a prospective study by Bartoletti et al. found that the odds of death in CAPA patients were 3.53 times the odds in the COVID-19 patients without IPA and that a decrease in mortality with antifungals was noted [9]. A study by White et al. showed significantly higher mortality rates in those with fungal diseases (53 vs. 32%; *p* = 0.04). Mortality was particularly high in those not receiving antifungal therapy (90% mortality) and significantly reduced (38.5%) in those who received antifungals (*p* = 0.008). However, a study by Hatzl et al. found that receiving antifungal prophylaxis upon admission did not improve survival in critically ill COVID-19 patients without any evidence of CAPA, despite a significant reduction in CAPA incidence [36]. In particular, 75 of 132 critically ill COVID-19 patients (57%) received antifungal prophylaxis, primarily posaconazole. Nine of the 10 patients later diagnosed with CAPA had not received prophylaxis. However, the study found no significant difference in 30-day mortality between the two groups, as the mortality was 37% in both groups [36]. These contradictory findings highlight the need for further research into antifungal prophylaxis against CAPA and treatment to better understand how to manage this serious complication. Moreover, antifungal prophylaxis and treatment practices may differ in various regions of the world, either of which could affect the incidence and outcome of CAPA. According to current guidelines, it is recommended to start treating CAPA as early as possible, even though this may result in some overtreatment and adverse drug events [13]. The preferred first-line drugs are voriconazole or isavuconazole, with liposomal amphotericin B (L-AMB) as the primary alternative [13]. In cases where azole resistance is suspected, voriconazole or isavuconazole can be combined with an echinocandin, and in cases of clear resistance, L-AMB should be preferred [13]. The prevalence of azole-resistant Aspergillus isolates can vary considerably depending on the geographic region. The prevalence of azole-resistant *Aspergillus fumigatus* varies significantly by geographic region [74]. The results from a multicenter study conducted in 19 European countries found that the prevalence of azole resistance was 3.2% and that the most common mutation was TR34/L98H [74]. Clinical samples from various countries have reported azole resistance rates of 2–12%, with Brazil, China, Japan, Pakistan, and the United States showing resistance rates ranging from 0.6 to 11.8% [75]. On the other hand, even higher rates of azole resistance have been reported for environmental samples, such as 13.9% in Tanzania and 9.3% in Colombia [75]. The emergence of azole-resistant environmental isolates is particularly concerning as they can serve as a potential reservoir for resistant strains that can then infect humans or animals, contributing to IPA treatment failure and increasing IPA mortality [76]. Therapeutic drug monitoring (TDM) is an essential tool that should be considered when using antifungal agents to treat patients with CAPA [13,69]. Antifungal agents such as voriconazole and isavuconazole have a narrow therapeutic window, meaning that their efficacy is highly dependent on achieving and maintaining optimal blood levels of the drug [77,78]. TDM involves monitoring drug levels in a patient’s blood to ensure that they remain within the therapeutic range, which can help optimize drug efficacy while minimizing the risk of toxic side effects [77]. For patients with CAPA, who may be critically ill and have multiple comorbidities, TDM can be particularly important, as it can help guide dosing adjustments and identify potential drug interactions [13,69,77]. Therefore, incorporating TDM into the management of CAPA patients receiving antifungal therapy can help improve treatment efficacy, minimizing adverse drug events [13,69,77].

In response to the emerging data indicating an increased risk of invasive fungal infections and mortality in critically ill patients with COVID-19, several countries have implemented screening protocols for these infections in severe COVID-19 patients admitted to ICUs [72,79]. Assessing the risk of fungal infection is particularly important for patients who have risk factors as per the Koehler et al. criteria, as well as those who receive prolonged or high-dose steroid treatment, individuals who require prolonged mechanical ventilation, and those with structural lung injury [72]. Identifying these risk factors can help healthcare providers take the necessary steps to prevent or promptly diagnose and treat fungal infections, which can be particularly dangerous for critically ill patients with COVID-19 [72]. If *Aspergillus* spp. are identified from respiratory samples of critically ill patients, they should be further evaluated [72]. Attributing such results to contamination or failing to acknowledge them as significant can be detrimental to these patients [72]. A study conducted by Ergün et al. established that a positive serum galactomannan (GM) test is the optimal predictor of mortality in patients diagnosed with CAPA [29]. The presence of *Aspergillus* growth detected in bronchoalveolar lavage (BAL) fluid culture, along with a positive result for BAL fluid GM, can serve as a strong predictor of 90-day mortality [33]. Individuals who test positive for both *Aspergillus* growth and GM in their BAL fluid are over two and a half times more likely to experience mortality within 90 days than those with both tests negative (HR, 2.53; 95% CI 1.28–5.02) [80]. Notably, this prediction holds even in the absence of a positive result for serum GM, suggesting that the BAL fluid culture and GM test may be particularly valuable in identifying high-risk individuals who might otherwise be missed [33]. Taken together, these findings underscore the importance of careful diagnostic evaluation in patients suspected of *Aspergillus* infection, as well as the need for prompt and targeted treatment in those who test positive for both *Aspergillus* growth and GM in their BAL fluid.

Microbiological diagnosis of IPA is critical in the management of patients with this condition, and prompt information is necessary to ensure proper treatment [72,80]. However, at the beginning of the pandemic, there were delays in laboratory testing, which may have led to delayed diagnoses and CAPA treatment. Therefore, it is essential that laboratories resume these diagnostic procedures as soon as possible to avoid any further delays and to ensure the best possible outcome for CAPA patients.

Our review has several limitations that should be acknowledged. First, the results presented in our review are only up to date as of December 2022, and there may be more recent studies or developments that were not included. Additionally, the diagnosis criteria for CAPA varied between studies, which could impact the accuracy and comparability of the results. Moreover, we were not able to include 30-, 42-, and 90-day mortality in the analysis of mortality due to the limited number of cases reported in the studies. Instead, only the number of deaths during ICU stays was considered (ICU mortality). While this approach provides some insight into the impact of CAPA on mortality, it may not fully capture the long-term effects of the complication. Although the reviewed articles reported the number of deaths for patients diagnosed with CAPA, it is important to note that only a limited number of articles included in this study presented separate data based on the classification distinguishing between possible and probable CAPA. Consequently, there is a lack of specific information on mortality rates for possible versus probable CAPA patients, which restricts our ability to directly compare the outcomes between these two groups. If the mortality was lower in possible CAPA patients compared to the probable CAPA patients, it could be argued that many possible CAPA cases could represent colonization instead of invasive disease. This limitation hampers our ability to draw any conclusions regarding the mortality differences between patients classified as possible and probable CAPA. Furthermore, the use of concomitant medications such as remdesivir and immunomodulatory drugs such as tocilizumab, baricitinib, vilobelimab, and glucocorticoids in the studies also played a crucial role in determining mortality. However, it is important to note that some of the studies reported incomplete information about these medications, such as the dosage, which could have affected the accuracy of our findings. Finally, it is important to recognize that the results of our review are not readily generalizable, as we did not include studies from all possible countries. Therefore, further research is needed to fully understand the characteristics and outcomes of CAPA patients across different regions and populations. Despite these limitations, our review provides valuable insights into the current state of knowledge on CAPA and highlights the need for continued research in this area.

## 5. Conclusions

In conclusion, the management of critically ill COVID-19 patients has significantly improved since the beginning of the pandemic, with a better understanding of the disease and the use of effective therapies. However, the risk of secondary infections, such as invasive fungal infections, remains a concern, especially among those with underlying medical conditions and those receiving corticosteroid therapy or mechanical ventilation. The newly defined clinical condition of CAPA has added to the challenges faced in the ICU management of COVID-19 patients. Further research and guidelines are needed to optimize the diagnosis and management of CAPA, as well as to improve the overall care of critically ill COVID-19 patients in ICUs.

## Figures and Tables

**Figure 1 jof-09-00689-f001:**
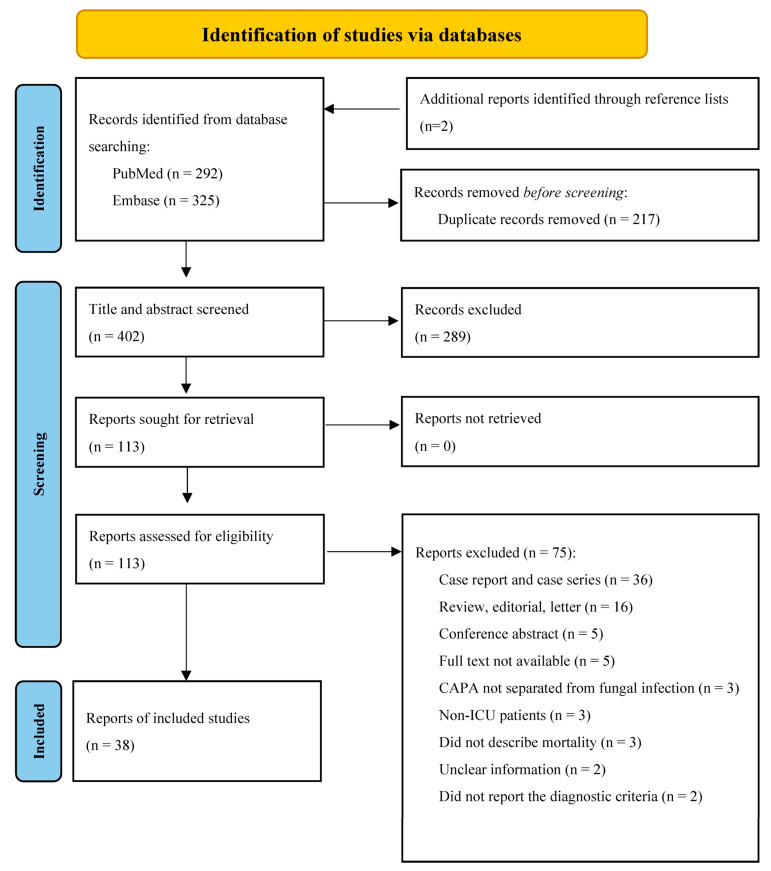
Database searches based on the PRISMA 2020 flow diagram.

**Figure 2 jof-09-00689-f002:**
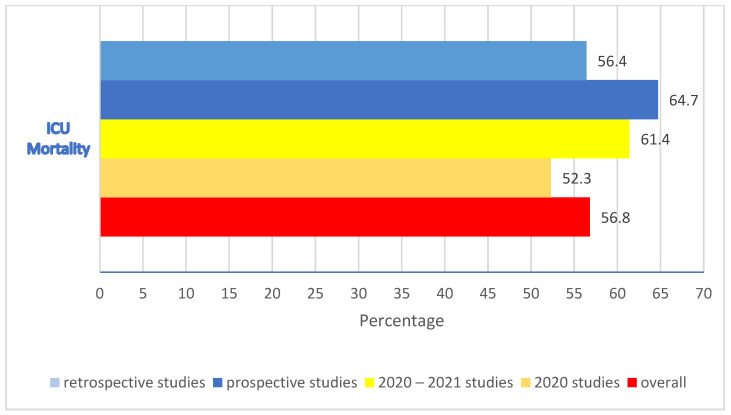
Overall ICU mortality in CAPA patients, stratified by study design and admission year.

## Data Availability

All analyzed data are included in this paper.

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
