# Peer review of "Mortality in ICU Patients with COVID-19-Associated Pulmonary Aspergillosis"

_jof, 2023, doi:10.3390/jof9060689_

Round 1
Reviewer 1 Report
Drs Beltrame and Haiduven provide a review on mortality in ICU patients with CAPA, and whether this has shifted throughout the pandemic. They should be commended to undertake this effort, which is very relevant as COVID-19 and CAPA are still frequent pathologies in the ICU. Moreover, their research question has to my knowledge never been investigated before. This manuscript will therefore definitively be an added value to the CAPA literature.
Major comment:
Given the controversy regarding the “possible” classification in the ECMM/ISHAM criteria, it would be interesting if the authors could calculate the mortality in those studies that used ECMM/ISHAM criteria comparing patients with possible versus probable CAPA. Indeed, if mortality would be low in the “possible” CAPA patients, this would be another argument to say that possible CAPA probably merely indicates colonization rather than invasive disease. Please add this calculation to the results.
Minor comments:
Introduction:
- First sentence: please adjust to “mainly caused by members of the ubiquitous Aspergillus fumigatus complex”, as for instance Aspergillus flavus, niger etcetera may also cause IPA.
Methods:
- Page 3, last sentence of first paragraph: the authors write “43-day mortality”, however in the last paragraph of the methods they write “42-day mortality”. Please correct, I presume they mean 42-day mortality.
- The authors state that they collected mortality (ICU-mortality, 30-day-, 42-day, or 90-day mortality). They do not list in-hospital mortality, while in the last paragraph they write that in-hospital mortality was a primary outcome parameter. This is a bit confusing, did all studies reporting 30-day, 42-day or 90-day mortality use explicitly state that they mean in-hospital mortality?
Results:
- What were the “other sources” from where the two other records were added?
- Line 180-182: the sum of the numbers of the criteria (30+3+4+2) is 39, while the authors included 38 studies. Is this a mistake?
- Line 188-189: the authors state that LOS ranged from 18 to 66 hours and length of ICU stay from 11 to 57 hours. This cannot be correct. Do they mean days? Moreover, do they mean median LOS?
Discussion
- The authors start the discussion with advancements in treatment of severe COVID-19. Corticosteroids, tocilizumab and remdesivir are discussed. However, to be complete, the authors should at least also discuss the implementation of baricitinib and vilobelimab. Please add.
- It would be good to add that a positive serum GM is the best predictor for mortality in CAPA patients, as was elegantly shown in the study by Ergün et al (J Clin Microbiol 2021).
- Please adjust in line 415-416: “Finally, it is important to recognize that the results of our review are not readily generalizable…”
References
- Please review the correctness of the references, for instance in reference 5 “Capa” should be “CAPA”
I have no comments on the quality of the English Language (except for a couple of typo's).
Author Response
Drs Beltrame and Haiduven provide a review on mortality in ICU patients with CAPA, and whether this has shifted throughout the pandemic. They should be commended to undertake this effort, which is very relevant as COVID-19 and CAPA are still frequent pathologies in the ICU. Moreover, their research question has to my knowledge never been investigated before. This manuscript will therefore definitively be an added value to the CAPA literature.
Author’s response: We thank the reviewer for these kind comments.
Major comment:
Given the controversy regarding the “possible” classification in the ECMM/ISHAM criteria, it would be interesting if the authors could calculate the mortality in those studies that used ECMM/ISHAM criteria comparing patients with possible versus probable CAPA. Indeed, if mortality would be low in the “possible” CAPA patients, this would be another argument to say that possible CAPA probably merely indicates colonization rather than invasive disease. Please add this calculation to the results.
Author’s response: I would like to express my gratitude to the reviewer for raising this important question. If the available mortality was lower in possible CAPA patients compared to the probable CAPA patients, that could argue that many possible CAPA cases could represent colonization instead of invasive disease. However, I regret to inform you that it was not possible to obtain the specific number of deaths for patients with possible CAPA versus probable CAPA in the available articles. Despite the articles reporting the number of deaths for the overall CAPA patients, only a limited number of articles presented separate data based on the classification of possible and probable CAPA.
I acknowledge that this limitation should have been addressed in the manuscript. This clarification is crucial for maintaining transparency and preventing any misinterpretation of the results. Thank you for bringing this to my attention.
We added these sentences to the limitation of the study. “Although the reviewed articles reported the number of deaths for patients diagnosed with CAPA, it is important to note that only a limited number of articles included in this study presented separate data based on the classification distinguishing between possible and probable CAPA. Consequently, there is a lack of specific information on mortality rates for possible versus probable CAPA patients, which restricts our ability to directly compare the outcomes between these two groups. If the mortality was lower in possible CAPA patients compared to the probable CAPA patients, that could argue that many possible CAPA cases could represent colonization instead of invasive disease. This limitation hampers our ability to draw any conclusions regarding the mortality differences between patients classified as possible and probable CAPA.”
Minor comments:
Introduction:
- First sentence: please adjust to “mainly caused by members of the ubiquitous Aspergillus fumigatus complex”, as for instance Aspergillus flavus, niger etcetera may also cause IPA.
Author’s response: Suggested addition made in line 28.
Methods:
- Page 3, last sentence of first paragraph: the authors write “43-day mortality”, however in the last paragraph of the methods they write “42-day mortality”. Please correct, I presume they mean 42-day mortality.
Author’s response: We meant 42-day mortality. Suggested addition made in line 112.
- The authors state that they collected mortality (ICU-mortality, 30-day-,42-day, or 90-day mortality). They do not list in-hospital mortality, while in the last paragraph they write that in-hospital mortality was a primary outcome parameter. This is a bit confusing, did all studies reporting 30-day, 42-day or 90-day mortality use explicitly state that they mean in-hospital mortality?
Authors’ response: Apologies for the confusion caused. We appreciate the reviewer for pointing out the discrepancy. We encountered difficulties in calculating the overall mortality due to variations in the criteria used across the included studies. Some studies reported ICU mortality, while others reported 30-day mortality, 42-day mortality, or 90-day mortality. As the most numerous data available were related to ICU mortality, we decided to focus on this criterion to obtain a pooled estimate of mortality. We have made the necessary corrections in the text in line 118.
Results:
- What were the “other sources” from where the two other records were added?
Author’s response: Apologies for the inaccuracy. The two additional records were obtained through a thorough examination of the reference lists of relevant articles. We have made the necessary revision in the revised manuscript to reflect this change accurately in lines 122-123.
- Line 180-182: the sum of the numbers of the criteria (30+3+4+2) is 39, while the authors included 38 studies. Is this a mistake?
Author’s response: Apologies for the error. The sum of the number of the criteria (30+3+4+1) have been properly changed in the revised manuscript in lines 196-197.
- Line 188-189: the authors state that LOS ranged from 18 to 66 hours and length of ICU stay from 11 to 57 hours. This cannot be correct. Do they mean days? Moreover, do they mean median LOS?
Author’s response: Apologies for the confusion caused by stating the LOS in hospital and ICU in hours instead of days. Considering variations in reporting (some articles used mean, others used median), we have chosen to remove the specific LOS range from the text in lines 202-203. However, the comprehensive range of median LOS for hospital and ICU, as well as the range of mean LOS for hospital and ICU, can be found in the corresponding table 2.
Discussion
- The authors start the discussion with advancements in treatment of severe COVID-19. Corticosteroids, tocilizumab and remdesivir are discussed. However, to be complete, the authors should at least also discuss the implementation of baricitinib and vilobelimab. Please add.
Author’s response: We thank the reviewer for pointing this out, & suggested addition made from lines 277 to 285 and in line 486.
- It would be good to add that a positive serum GM is the best predictor for mortality in CAPA patients, as was elegantly shown in the study by Ergün et al (J Clin Microbiol 2021).
Author’s response: We thank the reviewer for pointing this out, & suggested addition made from lines 441 to 443.
- Please adjust in line 415-416: “Finally, it is important to recognize that the results of our review are not readily generalizable…”
Author’s response: Suggested addition made in line 490.
References
Please review the correctness of the references, for instance in reference 5 “Capa” should be “CAPA”
Author’s response: We thank the reviewer for pointing this out, & suggested changes made.
Comments on the Quality of English Language
I have no comments on the quality of the English Language (except for a couple of typo's).
Author’s response: We thank the reviewer for this kind comment.
Reviewer 2 Report
In this paper, the authors reviewed 38 studies involving 1437 COVID patient admiited to ICU with CAPA to investigate whether mortality has changed since COVID pandemic. This work is well-designed and may serve a call to prioritize CAPA in clinic and research. However, some missing information is needed.
1. Line 115, “other sources” was cited in this paper. Please clarify these sources which may influence the interpretation of the results.
2. Since comorbidities such as chronic pulmonary diseases may increase the mortality risk, in Line 180,please present the comorbidities of these involved studies.
3. In Table 1, the sum percentage of proven, probable, putative and possible in some studies (such as ref 38, ref 40) seems not 100%. Please double-check the data.
4. In Table 2, the treatment regimens should contain more information. For instance, the dosage of dexamethasone which may impact the prognosis.
5. It is better some important information including the median mortality in prospective and retrospective studies, also present in table or figure.
Author Response
In this paper, the authors reviewed 38 studies involving 1437 COVID patients admitted to ICU with CAPA to investigate whether mortality has changed since COVID pandemic. This work is well-designed and may serve a call to prioritize CAPA in clinic and research.
Author’s response: We thank the reviewer for these kind comments.
However, some missing information is needed.
- Line 115, “other sources” was cited in this paper. Please clarify these sources which may influence the interpretation of the results.
Author’s response: Apologies for the inaccuracy. The two additional records were obtained through a thorough examination of the reference lists of relevant articles. We have revised the revised manuscript to reflect this change accurately in lines 122-123.
- Since comorbidities such as chronic pulmonary diseases may increase the mortality risk, in Line 180, please present the comorbidities of these involved studies.
Author’s response: We acknowledge the importance of considering comorbidities and their potential impact on mortality risk. We have incorporated a new table into the article to provide a comprehensive overview (Tab. 2).
- In Table 1, the sum percentage of proven, probable, putative and possible in some studies (such as ref 38, ref 40) seems not 100%. Please double-check the data.
Author’s response: Apologies for the error. The sum percentage of proven, probable, putative, and possible have been properly changed in the revised manuscript.
- In Table 2, the treatment regimens should contain more information. For instance, the dosage of dexamethasone which may impact the prognosis.
Author’s response: We sincerely apologize for the limitation in Table 2 regarding the treatment regimens and the absence of specific information, such as the dosage of dexamethasone. At the outset of this review, we made concerted efforts to collect comprehensive data on the treatment regimens, including the dosage of dexamethasone, used in the management of CAPA. Following your request, we conducted a new thorough review of all the articles in an attempt to obtain the dosage information for dexamethasone. However, despite our efforts, we were unable to acquire the necessary data regarding the dosage of dexamethasone from all the included studies.
We are aware that the lack of data regarding the dosage of dexamethasone is a limitation of our study, which may impact the interpretation of the findings. This limitation is discussed in the comment section of the manuscript (lines 488-489) to provide transparency and highlight the efforts made to address this issue.
- It is better some important information including the median mortality in prospective and retrospective studies, also present in table or figure.
Author’s response: We thank the reviewer for pointing this out. Fig. 2 has been added in lines 205, 207, and 210.
Round 2
Reviewer 1 Report
I thank the authors for their thorough responses to my comments. This is a valuable review that will definitely add to the existing literature. In my opinion the manuscript can be accepted for publication.
One optional remark though: I suppose Table 2 was added in response to another reviewer. In my opinion this six (!) pages long table is relevant, but not for the main text. I would add it to an online appendix/supplement and refer to it in the main text (otherwise pages 6 to 15 of the manuscript are tables). Also, adding the n-value of patients in each study in the first column would be helpful.
Reviewer 2 Report
No comments.